ecology

biotic interactions, colonization, priority effects, resistance and resilience

**Author for correspondence:**
Isabelle C. Barrett
e-mail: isabellecbarrett@gmail.com

# Negative resistance and resilience: biotic mechanisms underpin delayed biological recovery in stream restoration

Isabelle C. Barrett[1], Angus R. McIntosh[1], Catherine M. Febria[1,2] and Helen J. Warburton[1]

[1]School of Biological Sciences, University of Canterbury, Christchurch, New Zealand
[2]Great Lakes Institute for Environmental Research, University of Windsor, Windsor, Canada

ICB, 0000-0003-4132-3204

Traditionally, resistance and resilience are associated with good ecological health, often underpinning restoration goals. However, degraded ecosystems can also be highly resistant and resilient, making restoration difficult: degraded communities often become dominated by hyper-tolerant species, preventing recolonization and resulting in low biodiversity and poor ecosystem function. Using streams as a model, we undertook a mesocosm experiment to test if degraded community presence hindered biological recovery. We established 12 mesocosms, simulating physically healthy streams. Degraded invertebrate communities were established in half, mimicking the post-restoration scenario of physical recovery without biological recovery. We then introduced a healthy colonist community to all mesocosms, testing if degraded community presence influenced healthy community establishment. Colonists established less readily in degraded community mesocosms, with larger decreases in abundance of sensitive taxa, likely driven by biotic interactions rather than abiotic constraints. Resource depletion by the degraded community likely increased competition, driving priority effects. Colonists left by drifting, but also by accelerating development, reducing time to emergence but sacrificing larger body size. Since degraded community presence prevented colonist establishment, our experiment suggests successful restoration must address both abiotic and biotic factors, especially those that reinforce the 'negative' resistance and resilience which perpetuate degraded communities and are typically overlooked.

## 1. Introduction

Given widespread human impacts, restoration of degraded ecosystems is essential [1,2]. Terrestrial restoration work has generally focused on biotic components, predominantly plant communities [3,4], whereas addressing abiotic issues like soil legacies is less common [5]. By contrast, freshwater restoration tends to focus on physical habitat, water quality and other abiotic improvements under the assumption that improving habitat will directly enhance biodiversity and ecosystem function [6,7]. However, while physico-chemical improvements are often successful in aquatic systems, biological recovery, especially of macroinvertebrate communities, is less common [7–9]. In freshwater restoration, failure is often attributed to insufficient time since restoration [10]; however, even long-term post-restoration monitoring often fails to identify community recovery [11,12], indicating other factors are preventing recovery. Other impediments include incomplete physical recovery [7,13] or lack of colonization [14], typically associated with catchment-wide issues reducing the efficacy of reach-scale restoration [8,13]. Amelioration of these factors is essential, but unlike in terrestrial systems, the importance of

biotic interactions in aquatic community recovery has been largely ignored and is challenging to assess empirically [15,16]. Here, we present a mesocosm experiment simulating a post-restoration scenario in an aquatic system to investigate the mechanisms behind biotic recovery failure.

Environmental filters determine colonist survival in a given context, effectively defining their fundamental niche [17]. Given a regional species pool and no dispersal constraints, biotic interactions then determine successful establishment, defining species' realized niches [18]. In lotic systems, environmental degradation acting as an environmental filter selects for species with certain traits. For example, agricultural disturbance characterized by high nutrient and sediment loads often leads to domination by hardy, tolerant species such as those with protective cases (e.g. snails) and loss of more sensitive taxa such as Ephemeroptera (E, mayflies), Plecoptera (P, stoneflies) and Trichoptera (T, caddisflies) which are often notably sensitive to pollution [19]. Assuming complete physico-chemical restoration and an adequate colonist source, the remaining likely mechanisms of community assembly are biotic interactions, as demonstrated in terrestrial plant communities [20]. In particular, priority effects associated with degraded communities may be important for post-restoration community assembly. Order of arrival or 'priority effects' can influence coexistence and therefore community composition [21,22]. Species arriving early may reduce resource availability, thus inhibiting the survival of later arrivals (niche pre-emption), or may change the types of niches available to later arrivals, thus altering those able to establish (niche modification). From a restoration perspective, priority effects refer to the influences degraded communities, which established when conditions were poor and which persist after conditions improve, potentially have on the return of desired colonists. This parallels the inhibition model of succession in terrestrial plant communities [20]. Thus, in post-restoration scenarios where a pre-existing degraded community increases competition for space and resources, priority effects may prevent sensitive or specialist species from recolonizing.

Priority effects could mean a highly competitive degraded community dominated by hardy species will be stable and resistant to recolonization by desired species. This situation could parallel a hysteretic community state: hysteresis describes where a community is more easily shifted to one state than moved back [23,24]. In a restoration context, hysteresis suggests communities are more easily pushed into a degraded state than restored to a healthy one. Under restored environmental conditions, a degraded community may be highly resilient, maintaining its degraded state despite perturbations of environmental conditions. Therefore, priority effects associated with the pre-existing degraded community could underpin hysteresis in restoration [25].

Resistance and resilience are properties often associated with healthy ecosystems and communities alike; however, the context developed above suggests that degraded communities may also be inherently resistant and resilient to disturbances such as further environmental disturbance or even recolonization by other taxa. Resistance and resilience of *desired* communities are positive forces which preserve a healthy state, while resistance and resilience of *degraded* communities are negative forces which maintain degraded states [26]. Restoration aims to overcome the 'negative resistance and resilience' (*sensu* [26]) of the degraded state, facilitating the recovery of a positively resistant and resilient restored community.

We undertook a mesocosm experiment to investigate the consequences of negative resistance and resilience for biological recovery in aquatic restoration. We hypothesized that the presence of a persistent degraded community could inhibit desired colonist establishment, even if physico-chemical conditions are fully restored, predicting that in the presence of a degraded community, sensitive taxa (predefined based on existing data) would be lost from an ecosystem more quickly. We also investigated repercussions of degraded community presence on invertebrate development, predicting that development and subsequent emergence would be accelerated in sensitive taxa like mayflies when faced with a degraded community as they are when faced with predators [27].

## 2. Methods

Two restoration scenarios designed to mimic recolonization after abiotic recovery were established: desired colonist addition with an established degraded community present and desired colonist addition with no other invertebrates present as a control. Six replicate mesocosms for each treatment were established, arranged randomly in a four-by-three grid. The experiment ran for 42 days from April to June 2019, enabling investigation of colonist community change over time and sufficient to assess mayfly developmental changes; we would expect to see a significant change in size distribution for mayflies (*Deleatidium* spp.) over this time under typical healthy stream conditions [28].

Mesocosms mimicked physically and chemically restored streams with diverse flows and habitats (figure 1). Water pumped from Grasmere Stream, flowing through the University of Canterbury Cass Field Station bordering Arthur's Pass National Park, New Zealand (43°02'07.4" S 171°45'28.2" E), provided a consistent source of cool, oxygenated water and fine particulate organic matter [29]. Each mesocosm was circular, part flow-through and part recirculating; the outflow standpipe was positioned in a recess, reducing the rate at which invertebrates could leave via drift. A maximum velocity of $0.43 \pm SE\ 0.06\ m\ s^{-1}$ was established via three water jets (figure 1), with water movement and turnover maintaining temperature, dissolved oxygen and organic matter inputs (electronic supplementary material, appendix S1). The experiment was run in autumn, and as winter approached, temperature decreased, dissolved oxygen concentrations increased, pH became more alkaline and specific conductivity remained constant, but there were no significant differences between treatments (electronic supplementary material, appendix S1).

To enable the algal establishment and fine particulate organic matter to settle, mesocosms were turned on 7 days prior to the start of the experiment. After 4 days, degraded invertebrate communities were established in half of the 12 mesocosms, while the remaining six were left without invertebrates. The degraded communities were collected from a waterway with a macro-invertebrate fauna typical of agricultural drains, dominated by New Zealand mud snails, *Potamopyrgus antipodarum* which constituted 98% of community abundance. The remaining 2% comprised ostracods, *Xanthocnemis* damselflies, *Austrosimulium* blackfly larvae and chironomids. Invertebrates were measured into mesocosms by volume, achieving an initial density of approximately $27\,000\ m^{-2}$ (electronic supplementary material, appendix S2), consistent with degraded agricultural waterways in Canterbury, New Zealand [30].

On the first day of the experiment, desired colonist communities were added to all mesocosms. Recolonization under natural circumstances would be continuous, but this method enabled us to more easily identify taxa struggling to establish. Colonist communities comprised on average 89% EPT taxa, including 60% *Deleatidium* mayflies (electronic supplementary

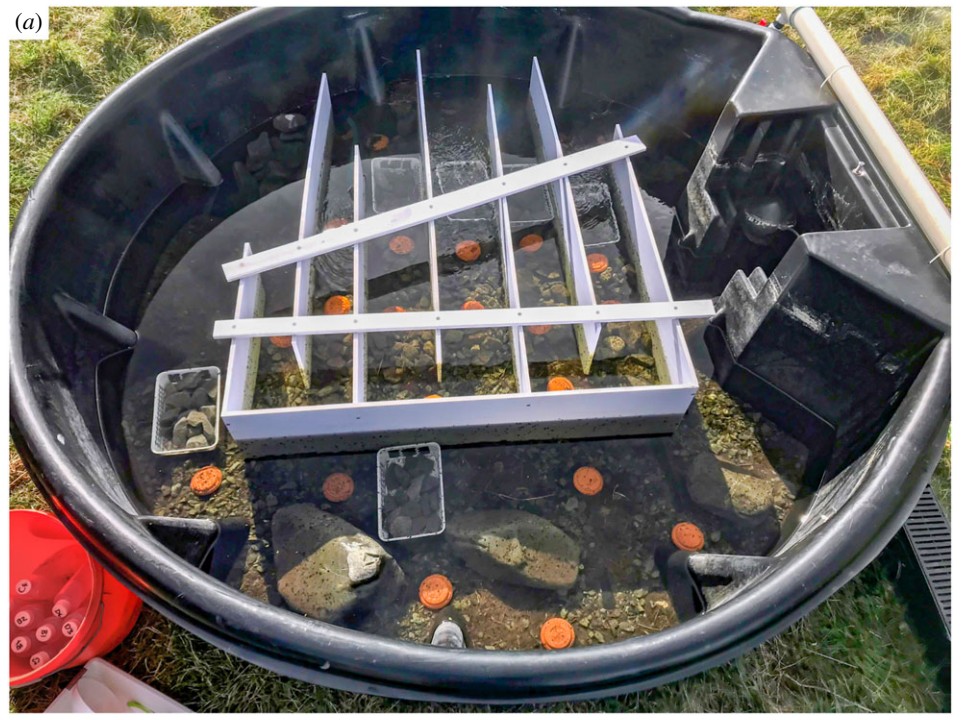

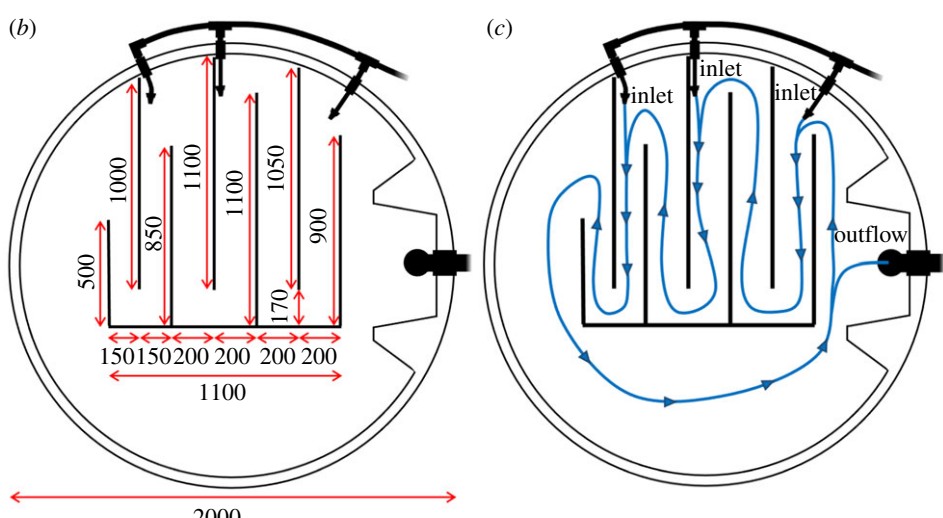

**Figure 1.** Design of mesocosms situated at the Cass field station in the Canterbury High Country, New Zealand showing (*a*) photo of baffle design and placement of rock baskets and algal tiles, (*b*) blueprint of mesocosm design including dimensions (mm) of mesocosms and baffles and (*c*) blueprint identifying three nozzle inlets (8 mm diameter), the location of the standpipe outlet (40 mm diameter) and resulting flow (blue line), directed by baffles. (Online version in colour.)

material, appendix S2). The remaining 11% included orders Mega-loptera (dobsonflies), Hemiptera (true bugs), Diptera (true flies) as well as snails and worms. This diverse assemblage is typical of healthy New Zealand streams [31]. Invertebrates were collected from local streams using 'electrobugging' [32]; we used a NIWA EFM300 electric fishing machine with a small, 19 cm electrode to produce a focused electric field, enabling us to catch large numbers of invertebrates with minimal physical damage to their bodies. Additional invertebrates, mainly caddisflies, were collected using gentle agitation of the benthos and kick nets. Invertebrates were transferred in aerated buckets. Each mesocosm received a similar colonist community using the 'benthic blender' method [33], whereby a 60 l colonist pool was consistently mixed using com-pressed air and identical sample volumes were extracted and sequentially assigned to each mesocosm until the invertebrate mix had been fully distributed. Once transferred to mesocosms, invertebrates were free to move around, including drifting out via the standpipe. Larvae and nymphs were also able to leave the system by emerging as adults. Thus, population reductions

could be attributed to emigration, emergence, mortality or preda-tion, although the degraded community contained relatively few predatory taxa besides the damselfly *Xanthocnemis zealandica*.

Six plastic baskets (16 × 23.5 cm), each containing 15 pebbles, were put into each mesocosm for use as invertebrate sampling units (figure 1a). Baskets were sampled on days 1, 6, 13 and 42 of the experiment. For the first three sampling occasions, three baskets per mesocosm were sampled: two from the riffle and one from the pool habitat to account for spatial variability, with different baskets sampled on consecutive sampling occasions to allow for recoloniza-tion. Invertebrates from each basket were identified and counted in the field before being returned to the mesocosms. Field identifi-cation likely meant some of the smallest invertebrates were missed, but replacing invertebrates with minimal disturbance was important to avoid depleting mesocosm communities. Commu-nities from these three sampling times were analysed separately to the final sample, where samples from all six baskets per mesocosm (preserved in 70% ethanol) were sorted and counted at 10–63× magnification, and identified based on Winterbourn *et al.* [34].

To correct for degraded community presence when assessing colonist community change, the dominant taxon in degraded communities, *P. antipodarum* snails, were excluded from community analysis. Other degraded community taxa were much less abundant, so the correction was not needed. To test if temporal changes in colonist community composition depended on treatment (i.e. presence of a degraded community), we used a combination of ordination and mixed-effects modelling. First, non-metric multidimensional scaling (NMDS) ordination, with square-root transformation and Wisconsin double standardization, of colonist communities from both treatments was conducted using *vegan* [35] in R [36] to assess temporal colonist community change. To test whether changes in communities depended on treatment, permutational multivariate analysis of variance (PERMANOVA [37]) was conducted with the *adonis* function in *vegan* [35] on Bray–Curtis distances from ordinations of community data.

To investigate how sensitive taxa fared in the presence of a degraded community, we calculated changes in the abundance of individuals belonging to EPT taxa. To give further insight, we also assessed counts of *Deleatidium* spp. mayflies and *P. antipodarum* snails, which were key taxa in the colonist and degraded communities respectively. To identify whether degraded community presence was responsible for changes in the aforementioned taxa counts over time, generalized linear modelling was performed for data from the first three sampling occasions with time as a continuous variable and each mesocosm as a replicate. In modelling, the quasipoisson distribution (log link) was used to deal with overdispersed data. The final samples from day 42 were analysed separately to account for different sampling methodology, using linear models to determine whether EPT, *Deleatidium* or *P. antipodarum* counts differed between control and degraded community treatments.

To assess degraded community influences on colonist life history, we again focused on *Deleatidium* genus mayflies. These are common across healthy New Zealand waterways [34] and were abundant in our colonist communities. *Deleatidium* development stage was assessed using wing bud development from the preserved samples. Individuals' total body length, excluding the cerci (following [38]), was measured (nearest millimetre) using an ocular micrometer. Wing bud development was scored 0–3 based on a standardized scheme (electronic supplementary material, appendix S2; [38]). A generalized linear model using the Poisson distribution (log link) was used to test if degraded community presence influenced *Deleatidium* body length at particular development stages. Emerged subimago mayflies resting on mesocosm edges were also counted during sampling on days 1 and 6. The poor flying ability of subimagos [39] meant that the movement of subimagos between mesocosms was unlikely. We tested how emergence varied with degraded community presence using a repeated-measures generalized linear mixed-effects model based on the Poisson distribution (log link) where treatment and time, the repeated measure, were fixed effects, mesocosm number was a random effect, and mesocosms were replicates.

To investigate degraded community influences on resource availability, algal biomass was measured using circular unglazed terracotta tiles (9.5 cm diameters; figure 1*a*) as substrate. Three tiles were sampled from each mesocosm 3 days before the experiment began and then again on days 1, 6, 13 and 42. Tiles were stored in the dark on ice for transportation to the laboratory where biofilm was removed with a toothbrush then filtered onto ashless filter papers (Whatman® 8 μm pore size). Filters were dried at 50°C for at least 24 h, weighed, ashed at 400°C for 2 h and weighed again to obtain ash-free dry mass (AFDM; [40]). The mean AFDM across the three tiles per mesocosm was then scaled up to determine algal biomass per square metre. For pre-experiment samples, a linear model was used to determine whether algal biomass in mesocosms destined for degraded communities differed from control mesocosms. For

samples taken during the experiment, a general linear model was used to test if changes in algal biomass over time differed between treatments. This included an interaction between time and treatment with time as a repeated measure.

## 3. Results

Degraded communities successfully established in mesocosms, and despite some drift out, initial rock basket samples show numbers stabilized at approximately 27 000 m$^{-2}$ prior to colonist community introductions (electronic supplementary material, appendix S2). Generalized linear modelling indicated that *P. antipodarum* abundance over the first three sampling occasions remained stable and low (less than 70 m$^{-2}$) in control mesocosms, but was high and gradually increasing in degraded community mesocosms ($F_{1,32} = 4.02$, $p = 0.05$; figure 2*a*). At the end of the experiment, *P. antipodarum* abundance in the degraded community had doubled, and was, unsurprisingly, much higher than in controls ($F_{1,8} = 172.5$, $p < 0.001$; figure 2*a*).

Over the first three time points, there was greater loss of EPT taxa and *Deleatidium* mayflies from colonist communities in the presence of a degraded community compared to the controls (EPT: $F_{1,32} = 32.26$, $p < 0.001$; figure 2*b*, *Deleatidium*: $F_{1,32} = 19.02$, $p < 0.001$; figure 2*c*). At the end of the experiment, there were fewer EPT taxa in colonist communities in the presence of a degraded community compared to the controls (EPT: $F_{1,8} = 4.94$, $p < 0.06$; figure 2*b*). This indicates faster and greater loss of sensitive taxa from colonist communities in degraded compared to control mesocosms. However, at the end of the experiment, there were actually more *Deleatidium* in colonist communities in the presence of a degraded community compared to the controls (*Deleatidium*: $F_{1,8} = 10.22$, $p < 0.05$; figure 2*c*).

Degraded community presence led to greater colonist community change over time, shown by movement further left along NMDS axis one (figure 3). For the first three time points, PERMANOVA identified a significant interaction between treatment and time ($F_{1,35} = 7.68$, $p < 0.001$) and a significant effect of treatment on community composition at day 42 ($F_{1,9} = 8.61$, $p < 0.01$), confirming divergence of colonist communities between treatments. Colonist community changes in the control mesocosms were driven by the loss of taxa across the board, whereas colonist community changes in the presence of a degraded community were driven by the loss of sensitive taxa including the caddisfly *Hydrobiosis*, the mayfly *Deleatidium* and stoneflies *Zelandoperla* and *Zelandobius*. The loss of these species was likely due predominantly to emergence or drift; however, some may be attributable to mortality (although this seems unlikely).

Generalized linear modelling of *Deleatidium* mayfly body length showed no interaction between development stage (characterized by wing development; electronic supplementary material, appendix S3) and treatment ($\chi^2_{2,410} = 1.10$, $p > 0.05$). However, there was a significant main effect of treatment, whereby *Deleatidium* were consistently smaller at each development stage in degraded mesocosms relative to control mesocosms, suggesting accelerated development in degraded mesocosms ($\chi^2_{1,410} = 10.01$, $p < 0.01$; figure 4*a*). As expected, there was also a significant main effect of development stage on length; more developed *Deleatidium* were longer in both degraded and control treatments ($\chi^2_{3,410} = 399.54$, $p < 0.001$;

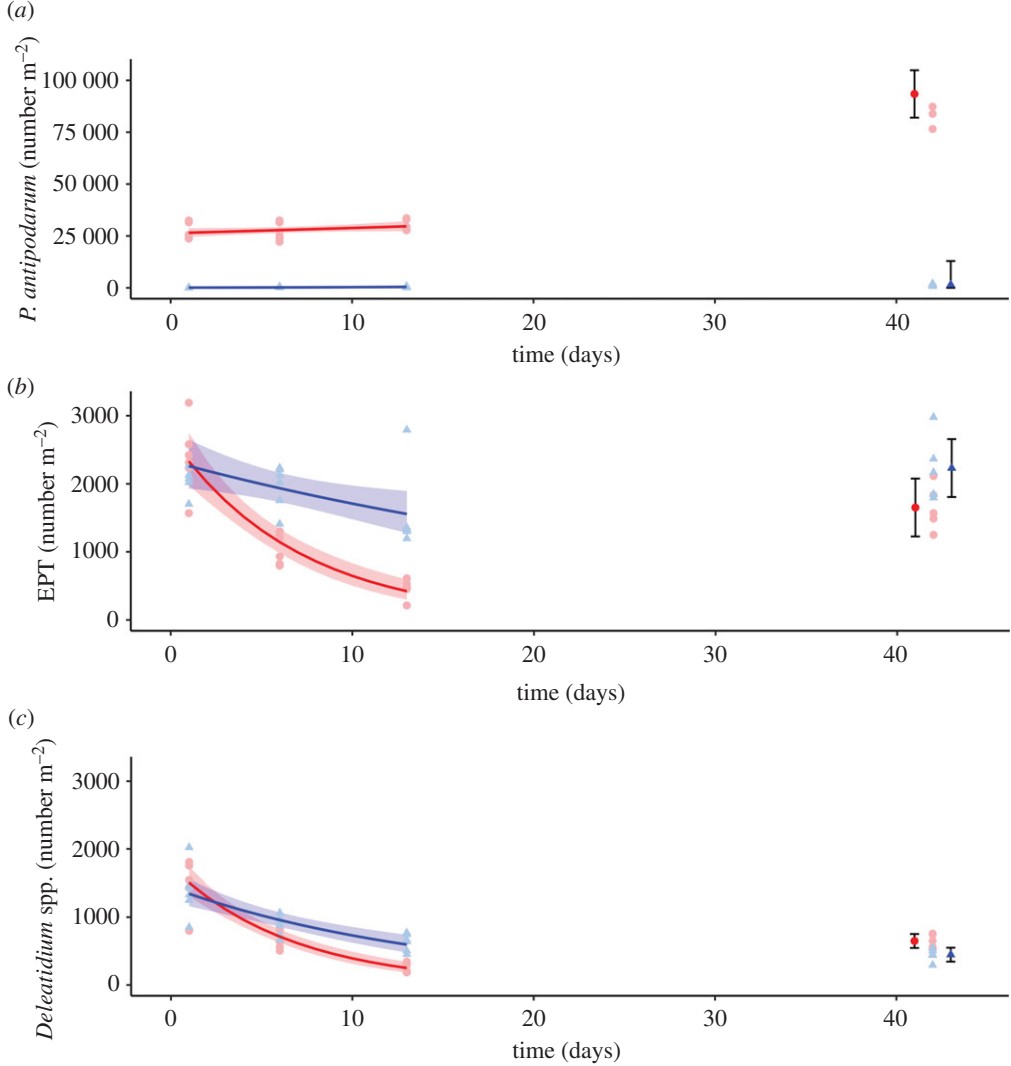

**Figure 2.** Changes in abundance of key macroinvertebrates over the first 13 days and at the end of the experiment in either the presence of a degraded community plus healthy colonists (red) or the empty control mesocosms plus healthy colonists (blue). Lines are model fits from GLMs over the first three time points, with shading representing 95% confidence intervals. Solid points are based on linear models of abundance at the end of the experiment, and error bars indicate standard error. Faded points indicate raw data for each mesocosm, based on combined samples from either three rock baskets per mesocosm (for days 1, 6 and 13) or six rock baskets per mesocosm (for day 42). *Potamopyrgus antipodarum* snails dominated the degraded communities (*a*). EPT counts (*b*) are the number of individuals from the generally pollution sensitive macroinvertebrate orders Ephemeroptera, Plecoptera and Trichoptera which were abundant in the colonist communities, and *Deleatidium* mayflies (*c*) were the most abundant species in the colonist communities. (Online version in colour.)

figure 4*a*). These observations were supported by mayfly emergence data; counts of emerged, subimago adults from mesocosm sides were significantly higher after 6 days than on day 1 ($\chi^2 = 95.46$, d.f. = 1, $p < 0.01$) and emergence was greater in degraded compared to control mesocosms ($\chi^2 = 96.39$, d.f. = 1, $p < 0.01$; figure 4*b*). The higher *Deleatidium* abundance in the presence of a degraded community compared to controls at the end of the experiment as mentioned previously (figure 2*c*) fits with these development and subsequent emergence patterns. By the end of the experiment, *Deleatidium* in the control mesocosms had likely completed growth and emerged (hence the low number of individuals remaining), while those in the presence of a degraded community which were not developed enough to emerge initially (even at a small size) likely suffered delays, resulting in a high abundance of smaller, underdeveloped individuals (figure 4*a*). It is likely many of these tiny, transparent individuals were missed in field counts on days 1, 6 and 13 and were only identified on day 42 with the help of a microscope.

There were no physico-chemical differences between treatments (electronic supplementary material, appendix S1), but the presence of a degraded community depleted algal resources compared to controls. There was no significant difference in AFDM between treatments prior to degraded community establishment ($F_{1,10} = 0.05$   $p > 0.05$); however, over the 3 days following degraded community establishment, and prior to the experiment beginning, algal biomass was reduced substantially (figure 5). During the experiment, algal biomass remained relatively constant ($F_{1,42} = 11.43$, $p < 0.01$), but lower in the presence of a degraded community than the controls ($F_{1,42} = 128.53$, $p < 0.001$), and there was no significant interaction between time and treatment ($F_{1,42} = 2.07$, $p > 0.05$).

## 4. Discussion

Freshwater restoration efforts can succeed in improving abiotic conditions [41,42], but there is often a lack of biological recovery and communities associated with degraded conditions

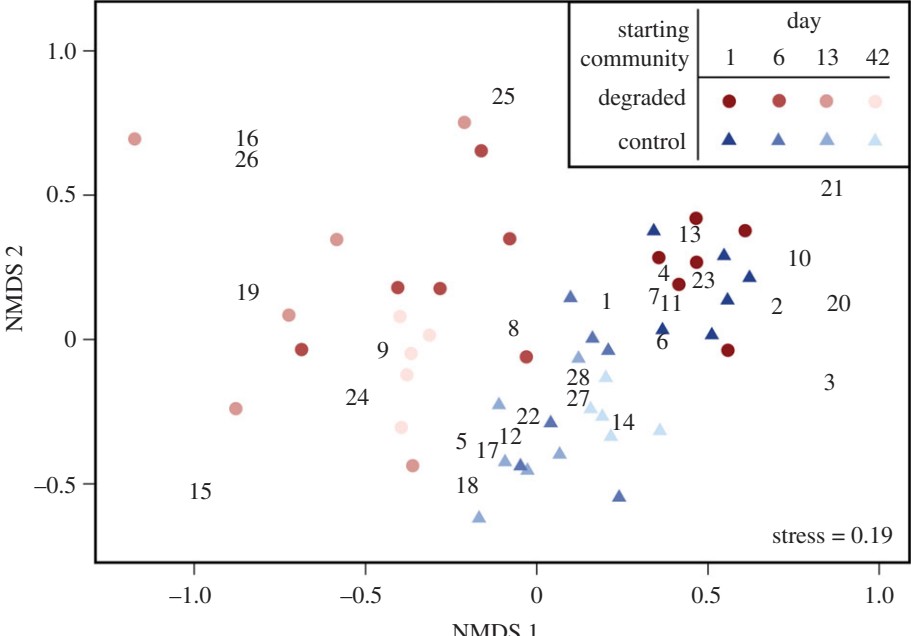

**Figure 3.** Non-metric multidimensional scaling (NMDS) ordination of colonist communities over time based on taxon abundances of colonist communities in mesocosms with ('degraded'; red) and without ('control'; blue) a pre-existing degraded community. Each point represents a macroinvertebrate community sampled from three rock baskets from one mesocosm, with colours becoming paler over time. Numbers represent the ordination of different taxa: *Aoteapsyche* (1), *Aphrophila* (2), *Archichauliodes* (3), *Austrosimulium* (4), *Beraeoptera* (5), Chironomidae (6), *Coloburiscus* (7), *Deleatidium* (8), *Hudsonema* (9), *Hydrobiosella* (10), *Hydrobiosis* (11), *Nesameletus* (12), Oligochaeta (13), *Olinga* (14), *Oniscigaster* (15), Ostracoda (16), *Oxyethira* (17), *Physa* (18), Turbellaria (19), *Polyplectropus* (20), *Psilochorema* (21), *Pycnocentria* (22), *Pycnocentrodes* (23), Sphaeriidae (24), *Triplectides* (25), *Xanthocnemis* (26), *Zelandobius* (27) and *Zelandoperla* (28). (Online version in colour.)

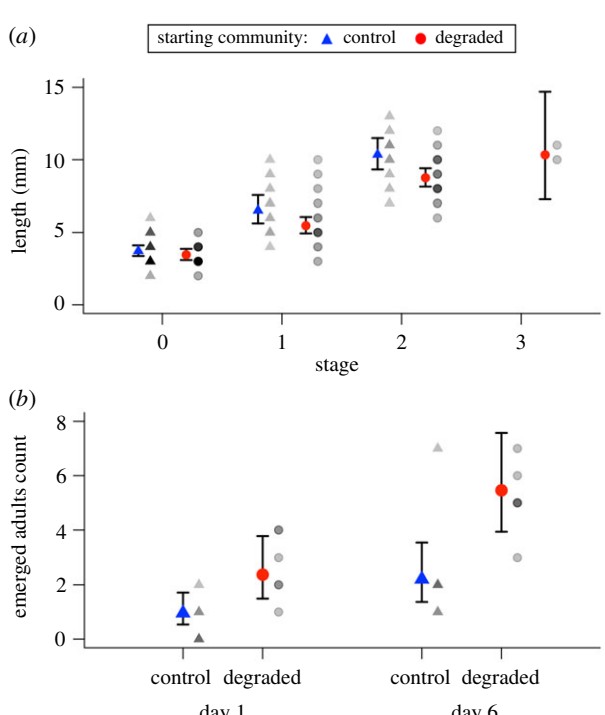

**Figure 4.** Relationship between development stage (described in electronic supplementary material, appendix S3: table A3) and mean body length (*a*) and patterns in emergence over time (*b*) of *Deleatidium* mayflies from mesocosms with a pre-existing degraded community plus healthy colonists (red) or empty controls with healthy colonists (blue). Error bars indicate modelled 95% confidence intervals. Grey points indicate raw data for each mesocosm, representing mean body length at each development stage per mesocosm (*a*), or counts of subimagos on the sides of mesocosms at the time of sampling (*b*), with darker points indicating overlapping data. (Online version in colour.)

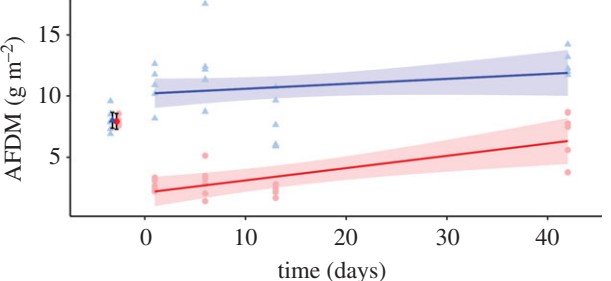

**Figure 5.** Algal biomass evaluated using ash-free dry mass (AFDM, grams per square metre) over time in the presence of a degraded community plus healthy colonists (red) and in the empty control mesocosms plus healthy colonists (blue). Solid points represent algal biomass prior to degraded community establishment, with standard error bars. Lines are based on a GLM of AFDM over time between treatments, with shaded sections showing 95% confidence intervals. Faded points indicate the average algal biomass across three tiles for each mesocosm. (Online version in colour.)

often persist [6,7,9,13]. Our results suggest the presence of these degraded communities can contribute to the lack of community recovery, likely inhibiting the establishment of sensitive taxa. In our experiment, colonist community change was far more pronounced in degraded community treatments, characterized by the loss of sensitive colonists. Colonization failure in the presence of a degraded community suggests that removal of abiotic stress is not enough to achieve community recovery if the degraded community persists, even when accompanied by colonist addition. Therefore, such priority effects in community assembly may underpin negative resistance and resilience (*sensu* [26]) and will need to be overcome to achieve successful, comprehensive restoration from degraded states. We outline

important aspects of these interactions below and summarize why such negative resistance and resilience is important.

We were interested in scenarios where negative resistance and resilience may arise, so our experiment focused on restoration contexts with an undesired community state characterized by hardy, tolerant taxa and excellent physico-chemical conditions. The degraded community was indeed very resistant to change in community composition, as shown by the increasing *P. antipodarum* abundance following colonist community addition. Our study may reflect *P. antipodarum*'s uniquely strong influence rather than a general degraded community effect, which will be important to investigate further. Nevertheless, this snail's prominence in degraded communities worldwide [43] means the negative resistance it drives will be important to address in other undesirable community states. Moreover, its effects serve to illustrate the general influences that degraded communities can have during restoration. Thus, although we don't know how an alternative pre-existing community might have affected our colonists, our experiment suggests priority effects are relevant to restoration and can render passive biotic rehabilitation measures ineffective. Thus, wherever community change is required, acknowledging existing communities and any historical contingencies, such as associated priority effects will be essential for successful restoration.

The idea that biotic interactions might influence, and potentially even outweigh, the effects of aquatic environmental filtering has been neglected but is not novel [16,21]. Our experiment shows that degraded community presence led to larger, more variable changes in colonist communities which became more apparent over time, suggesting that the degraded community hindered colonization. Changes to degraded mesocosm colonist communities were defined by the loss of sensitive taxa, including the caddisfly *Hydrobiosis*, the mayfly *Deleatidium*, and stoneflies *Zelandoperla* and *Zelandobius*. These taxa were lost at a greater rate in the presence of a degraded community than in the control mesocosms, reducing the likelihood of them successfully establishing. Similarly, decreasing colonist community EPT taxa over time in the presence of a degraded community indicated a reduction in colonist community health, suggesting the colonist community was moving to a state associated with degraded conditions despite excellent physical habitat and water quality. Thus, even with a good source of colonists and excellent physico-chemical conditions, recovery can be hindered by a persistent degraded community. Biotic interactions between colonists and the degraded fauna, particularly resource competition and possibly competition for space (both previously associated with *P. antipodarum* [44]), likely caused the differences in colonist establishment, and may underpin hysteresis in restoration. Therefore, more investigation of these biotic interactions in restoration contexts will also be important.

Resource depletion by the degraded community was likely a key driver of colonist community exclusion. Thus, under pressure from degraded community presence, characteristics of colonist community taxa, such as feeding habits or substrate attachment, were important in enabling them to persist under more intense resource competition, as indicated by previous studies of competition [45,46]. The dominant taxon in our degraded community, the snail *P. antipodarum*, is common in degraded communities across New Zealand and has become a pervasive invader globally [43]. Therefore, overcoming the negative resistance and resilience associated with

*P. antipodarum*-dominated communities will be important for stream restoration internationally. Populations of these snails are particularly good at exploiting scarce resources, for example, being able to graze algae almost down to the bare rock, thus reducing resources available to competing taxa [47,48]. Therefore, it was not surprising that *P. antipodarum* abundance increased substantially throughout the experiment, because these snails can attain very high densities even in the face of strong intraspecific competition [44,49]. Paradoxically, with good algal availability, *Deleatidium* mayflies, which dominated our colonist community, are more effective grazers than *P. antipodarum* [50]. Therefore, the observed persistence of the degraded community may be attributable to priority effects [51], whereby prior establishment facilitates higher abundances and therefore greater resource consumption. Although we do not know the exact mechanisms driving priority effects, ultimately, resource depletion by degraded community taxa may enable them to outcompete collectors and grazers such as *Deleatidium*, with repercussions throughout the food web [52].

Low resource levels via niche pre-emption may exacerbate priority effects [53,54], preventing later arrivals from establishing if other competitors have already taken hold; an effect observed in plant communities [53,55]. It is often hypothesized that the lack of colonization is responsible for poor post-restoration recovery [14], but our findings suggest that even with ample colonizers, an established degraded community could delay recovery. Moreover, resource competition and priority effects may underpin a negative feedback loop where hardy species persisting in a community are further boosted by improving conditions, effectively helping the 'rich get richer'. In our experiment, around a quarter of taxa found in the healthy colonist communities were also found in degraded communities (electronic supplementary material, appendix S2); taxa which were able to persist and boost the degraded communities. This may be a realization of facilitative priority effects of *P. antipodarum* on certain taxa, while other colonist taxa were inhibited [56]. This 'rich get richer' effect has already been noted in freshwater restoration, for example, Graham [57] found increasing detrital resources for EPT taxa led to a slight increase in EPT but a huge increase in *P. antipodarum* snails. Therefore, resource competition and priority effects are likely key mechanisms holding degraded communities in undesirable states, preventing colonists from gaining an initial foothold in the community and stagnating recovery. Our experiment only covered 42 days so we cannot directly infer long-term effects; however, results indicate serious short-term consequences which will delay recovery. Overcoming these will be essential for increasing restoration success and achieving tangible improvements sooner.

As well as short-term inability to successfully establish, interactions between degraded and colonist communities may have longer-term, more general impacts on population-level fitness of colonist populations which perpetuate the degraded state. Colonists under stress in our experiment could leave by drifting, dying or emerging. *Deleatidium* mayflies did the latter, but sacrificed growth, and therefore size, at emergence. Such effects on growth and development have been well documented in response to predatory fish [27,58]. For example, Peckarsky *et al*. (2001) found that *Baetis* mayfly nymphs developed faster but retained the same growth rate under predation stress resulting in adults with smaller body size and corresponding drops in fecundity. Our results suggest

biotic interactions within an invertebrate community undergoing restoration could have similar effects. Early emergence means individuals can escape unfavourable environments, but the associated life-history trade-off leading to a smaller size at emergence could reduce their fitness through reduced mating success and fecundity [59,60]. Reduced reproductive fitness of individuals can reduce population growth, potentially weakening the local colonist pool [61]. Not only would this process establish a negative feedback loop, further weakening colonization potential, but it will also perpetuate the negative feedback loop reinforcing stability of the established degraded community by further reducing the colonist pool.

Many of the processes described above reinforce the negative feedbacks which strengthen the dominance of existing communities in degraded systems, and likely underpin hysteresis. Our results show a degraded community can delay the desired colonist community from establishing, likely by negatively influencing sensitive taxa through competitive interactions, driven, for example, by resource depletion. From a restoration perspective, if biotic interactions are inciting negative resistance and resilience and preventing recovery, the restoration of biological communities will not be successful until the processes driving degraded community dominance are addressed. This would entail reaching a threshold (i.e. a tipping point) at which negative resistance and resilience are overcome and a community could move to a healthier state. Progress has been made in terrestrial plant restoration in acknowledging these ideas; negative resistance and resilience have been overcome by actively knocking back degraded communities [23,62] or manipulating competitive relationships through knowledge of priority effects to engender positive community change [4]. The need for a maintenance period of continued upkeep following restoration measures, like keeping weeds away [63], is commonly recognized in terrestrial restoration. Our study highlights the need for a similar approach to freshwater restoration. Investigating the mechanisms underpinning degraded community dominance and the application of biotic restoration methods to counter such problems will also be important. Fundamentally, the key to successful restoration across systems will be addressing both abiotic and biotic factors, acknowledging the role they both play in facilitating and inhibiting restoration.

Data accessibility. The dataset supporting this article is available on the BioHeritage National Science Challenge repository: (https://data.bioheritage.nz/dataset/edit/stream-mesocosm-restoration-experiment).

Authors' contributions. All authors conceived the ideas; I.C.B., A.R.M. and H.J.W. designed methodology; I.C.B. built the mesocosms, collected and analysed data, and led writing of the manuscript; all authors contributed critically to analysis and drafts and gave final approval for publication.

Competing interests. We declare we have no competing interests.

Funding. Our work, including support for I.C.B., H.J.W. and C.M.F., was funded by New Zealand's Ministry of Business, Innovation and Employment (through New Zealand's Biological Heritage National Science Challenge, C09X1501, Project 3.4, 'Reigniting Healthy Resilience'). C.M.F. and A.R.M. were also supported by a grant from the Mackenzie Charitable Foundation as part of the Canterbury Waterway Rehabilitation Experiment (CAREX).

Acknowledgements. We are grateful to Linda Morris for technical support, Ben McGinlay for assisting mesocosm design and construction, the University of Canterbury (UC) and Jenny Ladley for field station access, and UC's Freshwater Ecology Research Group for field help and feedback. In particular, we thank Kate Hornblow, Christopher Meijer, Alex Barclay, Alice West, Hayley Devlin, Gabriel Eve and Lochlan Boddy for field assistance. Discussions with or comments from Jon Harding, Kristy Hogsden, Elizabeth Graham, Simon Howard, Jonathan Tonkin, Marc Schallenberg and William Clements greatly improved the manuscript.

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
