## [Peer Review File · Proceedings of the Royal Society B: Biological Sciences]

Review History

RSPB-2020-1743.R0 (Original submission)

Review form: Reviewer 1

Recommendation

Accept with minor revision (please list in comments)

Scientific importance: Is the manuscript an original and important contribution to its field?

Excellent

General interest: Is the paper of sufficient general interest?

Excellent

Quality of the paper: Is the overall quality of the paper suitable?

Excellent

Is the length of the paper justified?

Yes

Should the paper be seen by a specialist statistical reviewer?

No

Do you have any concerns about statistical analyses in this paper? If so, please specify them explicitly in your report.

No

It is a condition of publication that authors make their supporting data, code and materials available - either as supplementary material or hosted in an external repository. Please rate, if applicable, the supporting data on the following criteria.

Is it accessible?

Yes

Is it clear?

N/A

Is it adequate?

N/A

Do you have any ethical concerns with this paper?

No

Comments to the Author

This manuscript points out that while many studies and discussions seek to promote ecosystem resiliency, degraded ecosystems can also be resistant or resilient in ways that prevent their recovery to previous, undisturbed states. While environmental filtering is an important determinant of community composition and structure, and can be a barrier to restoration, this study instead looks at the effect of biotic interactions on recolonization of restored habitat. Overall, their finding is that the presence of species that were adapted to degraded conditions can hinder community recovery even once conditions improve. This is an important message. This manuscript was well-written and an enjoyable read – I appreciated how the authors connected this very practical question (how to restore communities) with ecological theory. The discussion of potential longer-term consequences through effects on fitness is also very nice.

Having said that, I think that the paper would benefit from a more clear definition of priority effects. Simply being there first does not imply that the reason a species/guild/community can persist is due to priority effects – they may be competitively superior anyway, in which case it wouldn't matter if they had arrived first. Since *P. antipodarum* is an invasive species worldwide, could it establish even in healthy stream communities in the study area? There are several ways to define priority effects – two through niche theory (niche preemption or niche modification), and one through coexistence theory (that interspecific competition is stronger than intraspecific competition, so whichever competitor has the higher abundance will have the upperhand). I could imagine any of these three mechanisms being true for the authors' study system (the snails could certainly modify available niches, and L340-341 suggests that they may be fairly impervious to intraspecific competition). It is unfortunate that the authors did not have a treatment with high abundance of the "original" community and low/medium abundance of the "degraded" community, or some other combination of relative densities – this would be the way to establish whether a priority effect is truly occurring. Regardless, the authors should be more clear about exactly what they mean by priority effects, if they wish to use it as a central explanation of their findings.

My other major comment is that this paper is framed (including in the title) as being about resistance and resilience, yet does not perform any of the analyses typically associated with these questions. By excluding the "degraded" community from the ordination analyses, the authors don't really look at whether they are resistant/resilient to the colonizers' arrival. Instead of characterizing the total community, most of the analyses are about the colonizer community. This is definitely interesting, and required in order to understand the mechanisms happening. However, it is just a bit of a surprise given the main framing of the paper. This could be

addressed by making some more specific links in the discussion – for example, somewhere in the discussion, maybe around lines 301-302, say something like, “The degraded community was indeed very resistant to change in community composition, as shown by the maintenance and even increase of *P. antipodarum* abundance after the addition of the colonist community.”

Addressing these issues would require just some editing and potentially a slight reframing of parts of the introduction and discussion. This is an excellent study and manuscript, and with some clarification it will be an important contribution to both conceptual and applied ecology literature. It is also very nice to see the authors bringing ideas that are familiar in terrestrial settings into the aquatic realm; such connections are always beneficial.

I have few other comments. The description of the experiment is thorough and the design seems well-thought-out, and statistical analyses appear to be appropriate.

L305-306: As mentioned in the general comment above, this experiment doesn't exactly prove that priority effects are the mechanism through which this is occurring. Changing wording to “suggests” could be better.

L306-308: Likewise, here I would suggest changing the wording to “existing communities and any historical contingencies” would be more appropriate.

Review form: Reviewer 2

Recommendation

Accept with minor revision (please list in comments)

Scientific importance: Is the manuscript an original and important contribution to its field?
Excellent

General interest: Is the paper of sufficient general interest?
Excellent

Quality of the paper: Is the overall quality of the paper suitable?
Excellent

Is the length of the paper justified?
Yes

Should the paper be seen by a specialist statistical reviewer?
No

Do you have any concerns about statistical analyses in this paper? If so, please specify them explicitly in your report.
No

It is a condition of publication that authors make their supporting data, code and materials available - either as supplementary material or hosted in an external repository. Please rate, if applicable, the supporting data on the following criteria.

Is it accessible?
Yes

Is it clear?

Yes

Is it adequate?

Yes

Do you have any ethical concerns with this paper?

No

Comments to the Author

Title

I suggest to add a word like “aquatic” or “stream” to the title to underpin that the study was conducted in the aquatic environment and focused on stream restoration.

Abstract

L20-23. This sentence, definition and prove of the very important term “negative resistance and resilience” is one of the main conclusions of the paper, thus it would fit much better at the end of the abstract as a conclusion rather than as the second sentence.

Introduction

L68 I think it is important to mention that this reference refers to terrestrial plant communities and it is not one on one transferable to aquatic flowing (not standing) systems.

L79ff This is fine for me, but isn't there a time factor inherent to the “succession”?

L105 the accessory sentence could be deleted, as the species are not lost to the community in general if they emerge, they are only lost to the current aquatic spot.

Methods

L181 asses needs one more “s”

L208ff The authors need to explain how they ensured that the counted subimagines emerged from the specific mesocosm and not from an adjacent one of a potential different treatment.

Why weren't the imagines counts also done in week 2 (day 13) and the end (42)?

Results

L238ff The text jumps between results of figure 2 and 3. I suggest to first explain figure 2 and then add results of figure 3.

The explained variance of the two NMDS axes need to be added, to understand their influence.

L243 the authors should add if the loss of those species was due to natural emergence, drift, predation or other factors

Pool and riffle samples of each sample date were kept separately (see methods), so the results should be indicated or at least indicated if there were no significant differences.

Discussion

L297 “is” important

L305/306 please re-write. The conclusion is too straight forward and does not account for any time frame longer than 13 days. Even the results of the end day of the experiment show that there will be some succession, not to mention potential changes in upcoming years or decades.

L307 “likely” can be deleted

L327 Just a comment: particularly on the long run – how long will this negative resistance remain, will be one of the next questions to be answered

L331ff this is true for the grazers and collectors but what about the predators?

L347 again I think this cannot be stated in an absolute manner; there are no data yet that recovery is completely prevented (only 13/42 days). Recovery is probably delayed and most probably delayed for years but not forever.

L378 impede or delay or hinder is in the sense above better than “prevent”

L382ff this is a very important point and might also be stated in the abstract

Figures

Fig. 2. Normally the results of the end day should be displayed together with the results of the first three samplings in one diagram. Thus, B, D and F is not necessary. The curves probably do not look so nice because they increase again but the message is still the same and the interpretation in the result section can remain.

Fig 4. It should be added why there are no counts of stage 3 in Fig 4 A.

I am not sure but someone has to decide if the plural of subimago is subimagine, subimagos or subimagoes. The latter one is wrong I think.

Decision letter (RSPB-2020-1743.R0)

19-Aug-2020

Dear Miss Barrett:

I am writing to inform you that your manuscript RSPB-2020-1743 entitled "Negative resistance and resilience: biotic mechanisms underpin delayed biological recovery in restoration" has, in its current form, been rejected for publication in Proceedings B.

This action has been taken on the advice of referees, who have recommended that substantial revisions are necessary. With this in mind we would be happy to consider a resubmission, provided the comments of the referees are fully addressed. However please note that this is not a provisional acceptance.

Sincerely,

Professor Hans Heesterbeek

Associate Editor

Board Member: 1

Comments to Author:

I have now received two expert reviews of this manuscript, and have carefully read it again myself in light of those reviews. The reviewers and I share enthusiasm for the fact that the manuscript connects an important question in applied ecology with basic ecological principles, and feel that it could become an impactful contribution. The fundamental message of the paper – that biotic interactions may create obstacles to successful stream restoration even when abiotic conditions have improved – seems like an important one for the large and global community interested in stream restoration, and like it has at least some relevance to terrestrial restoration ecologists as well.

The reviewers highlight several important concerns about the extent to which the study supports some of the terms and concepts used to describe it. I summarize these concerns below, along with some similar additional concerns of my own. A revised manuscript that addressed these concerns thoughtfully and thoroughly would be a much stronger contribution to the literature despite – or because of – being a little more cautious and introspective, especially if it could simultaneously retain the kind of clear narrative that both the reviewers and I found so enjoyable and thought-provoking to read.

First, it is not clear whether the observed differences between treatments in colonization success for the “healthy” community were due to the presence of an established “_degraded_” community, or simply to the presence of _an_ established community. The manuscript acknowledges this point in the paragraph at line 299, and makes a reasonable case for focusing on a situation where a degraded community has already established. Nonetheless, the discussion here felt a bit superficial. I was left wondering what the community ecology literature says about colonization or invasion of established communities, and how to interpret the results here in light of that literature. As one small piece of a discussion of this issue, it might be valuable to consider the role of disturbance in initiating new succession sequences; could something like a flood disturb a degraded community to the extent that it then became invasible by the healthy community?

Similarly, if it is in fact the degraded community that is important, it is not clear to what extent the results were attributable to the presence of a degraded community in general, rather than to the presence of *P. antipodarum* in particular. My understanding is that this species is a particularly successful and impactful invader of stream ecosystems, including perhaps those where abiotic conditions have not been degraded. The reader wonders to what extent the results represent the uniquely strong effects of this species, rather than a generalizable effect of degraded communities.

Reviewer 1 points out that the degraded community colonist, *P. antipodarum*, might outcompete the healthy colonist community regardless of the order in which the two arrive. This hypothesis cannot be evaluated given the study design, of course, but it does seem plausible given the success of *P. antipodarum* as an invader, and it calls into question the framing of the results in terms of priority effects. Fortunately, even in this case the paper seems significant, because the message that biotic interactions can hinder restoration goals even when abiotic conditions are restored seems like an important and novel one, and it still stands. Nonetheless, the reviewer’s comment emphasizes the need to better justify, tone down, or abandon the framing in terms of priority effects. If you do retain this framing, the reviewer offers some important suggestions about defining this concept more clearly and with a stronger link to existing literature.

Both reviewers also raise critiques of your use of the resistance and resilience concepts, highlighting some ways in which the manuscript falls short of fully engaging with these ideas.

Beyond these major conceptual concerns, the reviewers raise a number of suggestions and questions which should be addressed in a revised manuscript. Some of these bear directly on the strength of the data; for instance, Reviewer 2 asks whether you were able to accurately determine that subimagos had emerged from the mesocosm where you found them resting, rather than from another mesocosm. To their detailed comments I'll add one of my own: I was a little confused by the *Deleatidium* results (paragraph at line 258): how is it that high abundance of this species in the degraded treatment at the end of the experiment is consistent with higher emergence of this species from the degraded treatment?

Reviewer(s)' Comments to Author:

Referee: 1

Comments to the Author(s)

This manuscript points out that while many studies and discussions seek to promote ecosystem resiliency, degraded ecosystems can also be resistant or resilient in ways that prevent their recovery to previous, undisturbed states. While environmental filtering is an important determinant of community composition and structure, and can be a barrier to restoration, this study instead looks at the effect of biotic interactions on recolonization of restored habitat. Overall, their finding is that the presence of species that were adapted to degraded conditions can hinder community recovery even once conditions improve. This is an important message. This manuscript was well-written and an enjoyable read – I appreciated how the authors connected this very practical question (how to restore communities) with ecological theory. The discussion of potential longer-term consequences through effects on fitness is also very nice.

Having said that, I think that the paper would benefit from a more clear definition of priority effects. Simply being there first does not imply that the reason a species/guild/community can persist is due to priority effects – they may be competitively superior anyway, in which case it wouldn't matter if they had arrived first. Since *P. antipodarum* is an invasive species worldwide, could it establish even in healthy stream communities in the study area? There are several ways to define priority effects – two through niche theory (niche preemption or niche modification), and one through coexistence theory (that interspecific competition is stronger than intraspecific competition, so whichever competitor has the higher abundance will have the upperhand). I could imagine any of these three mechanisms being true for the authors' study system (the snails could certainly modify available niches, and L340-341 suggests that they may be fairly impervious to intraspecific competition). It is unfortunate that the authors did not have a treatment with high abundance of the "original" community and low/medium abundance of the "degraded" community, or some other combination of relative densities – this would be the way to establish whether a priority effect is truly occurring. Regardless, the authors should be more clear about exactly what they mean by priority effects, if they wish to use it as a central explanation of their findings.

My other major comment is that this paper is framed (including in the title) as being about resistance and resilience, yet does not perform any of the analyses typically associated with these questions. By excluding the "degraded" community from the ordination analyses, the authors don't really look at whether they are resistant/resilient to the colonizers' arrival. Instead of characterizing the total community, most of the analyses are about the colonizer community. This is definitely interesting, and required in order to understand the mechanisms happening. However, it is just a bit of a surprise given the main framing of the paper. This could be addressed by making some more specific links in the discussion – for example, somewhere in the discussion, maybe around lines 301-302, say something like, "The degraded community was indeed very resistant to change in community composition, as shown by the maintenance and even increase of *P. antipodarum* abundance after the addition of the colonist community."

Addressing these issues would require just some editing and potentially a slight reframing of parts of the introduction and discussion. This is an excellent study and manuscript, and with

some clarification it will be an important contribution to both conceptual and applied ecology literature. It is also very nice to see the authors bringing ideas that are familiar in terrestrial settings into the aquatic realm; such connections are always beneficial.

I have few other comments. The description of the experiment is thorough and the design seems well-thought-out, and statistical analyses appear to be appropriate.

L305-306: As mentioned in the general comment above, this experiment doesn't exactly prove that priority effects are the mechanism through which this is occurring. Changing wording to "suggests" could be better.

L306-308: Likewise, here I would suggest changing the wording to "existing communities and any historical contingencies" would be more appropriate.

Referee: 2

Comments to the Author(s)

Title

I suggest to add a word like "aquatic" or "stream" to the title to underpin that the study was conducted in the aquatic environment and focused on stream restoration.

Abstract

L20-23. This sentence, definition and prove of the very important term "negative resistance and resilience" is one of the main conclusions of the paper, thus it would fit much better at the end of the abstract as a conclusion rather than as the second sentence.

Introduction

L68 I think it is important to mention that this reference refers to terrestrial plant communities and it is not one on one transferable to aquatic flowing (not standing) systems.

L79ff This is fine for me, but isn't there a time factor inherent to the "succession"?

L105 the accessory sentence could be deleted, as the species are not lost to the community in general if they emerge, they are only lost to the current aquatic spot.

Methods

L181 asses needs one more "s"

L208ff The authors need to explain how they ensured that the counted subimagines emerged from the specific mesocosm and not from an adjacent one of a potential different treatment.

Why weren't the imagines counts also done in week 2 (day 13) and the end (42)?

Results

L238ff The text jumps between results of figure 2 and 3. I suggest to first explain figure 2 and then add results of figure 3.

The explained variance of the two NMDS axes need to be added, to understand their influence.

L243 the authors should add if the loss of those species was due to natural emergence, drift, predation or other factors

Pool and riffle samples of each sample date were kept separately (see methods), so the results should be indicated or at least indicated if there were no significant differences.

Discussion

L297 "is" important

L305/306 please re-write. The conclusion is too straight forward and does not account for any time frame longer than 13 days. Even the results of the end day of the experiment show that there will be some succession, not to mention potential changes in upcoming years or decades.

L307 "likely" can be deleted

L327 Just a comment: particularly on the long run – how long will this negative resistance remain, will be one of the next questions to be answered

L331ff this is true for the grazers and collectors but what about the predators?

L347 again I think this cannot be stated in an absolute manner; there are no data yet that recovery is completely prevented (only 13/42 days). Recovery is probably delayed and most probably delayed for years but not forever.

L378 impede or delay or hinder is in the sense above better than “prevent”

L382ff this is a very important point and might also be stated in the abstract

Figures

Fig. 2. Normally the results of the end day should be displayed together with the results of the first three samplings in one diagram. Thus, B, D and F is not necessary. The curves probably do not look so nice because they increase again but the message is still the same and the interpretation in the result section can remain.

Fig 4. It should be added why there are no counts of stage 3 in Fig 4 A.

I am not sure but someone has to decide if the plural of subimago is subimagines, subimagos or subimagos. The latter one is wrong I think.

Author's Response to Decision Letter for (RSPB-2020-1743.R0)

See Appendix A.

RSPB-2021-0354.R0

Review form: Reviewer 1

Recommendation

Accept as is

Scientific importance: Is the manuscript an original and important contribution to its field?

Excellent

General interest: Is the paper of sufficient general interest?

Good

Quality of the paper: Is the overall quality of the paper suitable?

Excellent

Is the length of the paper justified?

Yes

Should the paper be seen by a specialist statistical reviewer?

No

Do you have any concerns about statistical analyses in this paper? If so, please specify them explicitly in your report.

No

It is a condition of publication that authors make their supporting data, code and materials available - either as supplementary material or hosted in an external repository. Please rate, if applicable, the supporting data on the following criteria.

Is it accessible?

Yes

Is it clear?

Yes

Is it adequate?

Yes

Do you have any ethical concerns with this paper?

No

Comments to the Author

This manuscript describes a well-designed study to investigate the effects of biotic interactions on the establishment of natural invertebrate communities to restored stream habitats. The authors comprehensively addressed most comments from the previous reviews, adding precision and nuance to the introduction and discussion as well as explaining a few methodological issues. I consider this to be a strong and well-written manuscript and have no major comments.

Minor comments:

L179 and Figure 3: The methods text states that the snails were left out of community analyses, but the NMDS plots show species 21 as *Potamopyrgus*. This is a little confusing.

Figure 2: I can see from the documents included that Figure 2 was edited according to suggestions by a previous reviewer. However, I find this presentation with a smooth line for days 1-13 and then some points at day 42 to be very jarring. I don't consider it to be an improvement on the previous Figure 2, but it is ultimately up to the authors' preference!

Review form: Reviewer 2

Recommendation

Accept as is

Scientific importance: Is the manuscript an original and important contribution to its field?

Excellent

General interest: Is the paper of sufficient general interest?

Excellent

Quality of the paper: Is the overall quality of the paper suitable?

Excellent

Is the length of the paper justified?

Yes

Should the paper be seen by a specialist statistical reviewer?

No

Do you have any concerns about statistical analyses in this paper? If so, please specify them explicitly in your report.

No

It is a condition of publication that authors make their supporting data, code and materials available - either as supplementary material or hosted in an external repository. Please rate, if applicable, the supporting data on the following criteria.

Is it accessible?

Yes

Is it clear?

Yes

Is it adequate?

Yes

Do you have any ethical concerns with this paper?

No

Comments to the Author

After all the very good improvements I have only one small comment: in Figure 3 the species number 29 is missing in the figure. This should be added.

Decision letter (RSPB-2021-0354.R0)

02-Mar-2021

Dear Miss Barrett

I am pleased to inform you that your manuscript RSPB-2021-0354 entitled "Negative resistance and resilience: biotic mechanisms underpin delayed biological recovery in stream restoration" has been accepted for publication in Proceedings B.

The referees have recommended publication, but also suggest some minor revisions to your manuscript. Therefore, I invite you to respond to the referees' comments and revise your manuscript. Because the schedule for publication is very tight, it is a condition of publication that you submit the revised version of your manuscript within 7 days. If you do not think you will be able to meet this date please let us know.

Sincerely,
Professor Hans Heesterbeek
mailto: proceedingsb@royalsociety.org

Associate Editor
Comments to Author:

Thank you for your careful attention to revising the manuscript and responding to the reviews. The manuscript is now close to being acceptable for publication in Proc B. Please attend to the few remaining minor issues highlighted in the reviewers' comments. Also, please make sure that the data are fully accessible - one of the reviewers noted that, while they were able to find the data via the BioHeritage website, the link provided in the paper didn't work for them.

Please address these issues expeditiously by submitting a revised version. This revision will not be sent out for further review, but will be checked by our editorial staff to make sure that the paper is suitable to be sent to production.

Thank you again for submitting this interesting work to Proc B, we look forward to seeing it in print.

Reviewer(s)' Comments to Author:

Referee: 1

Comments to the Author(s).

This manuscript describes a well-designed study to investigate the effects of biotic interactions on the establishment of natural invertebrate communities to restored stream habitats. The authors comprehensively addressed most comments from the previous reviews, adding precision and nuance to the introduction and discussion as well as explaining a few methodological issues. I consider this to be a strong and well-written manuscript and have no major comments.

Minor comments:

L179 and Figure 3: The methods text states that the snails were left out of community analyses, but the NMDS plots show species 21 as *Potamopyrgus*. This is a little confusing.

Figure 2: I can see from the documents included that Figure 2 was edited according to suggestions by a previous reviewer. However, I find this presentation with a smooth line for days 1-13 and then some points at day 42 to be very jarring. I don't consider it to be an improvement on the previous Figure 2, but it is ultimately up to the authors' preference!

Referee: 2

Comments to the Author(s).

After all the very good improvements I have only one small comment: in Figure 3 the species number 29 is missing in the figure. This should be added.

Author's Response to Decision Letter for (RSPB-2021-0354.R0)

See Appendix B.

Decision letter (RSPB-2021-0354.R1)

08-Mar-2021

Dear Miss Barrett

I am pleased to inform you that your manuscript entitled "Negative resistance and resilience: biotic mechanisms underpin delayed biological recovery in stream restoration" has been accepted for publication in Proceedings B.

Open Access

Paper charges

Sincerely,

Proceedings B

Appendix A

First, it is not clear whether the observed differences between treatments in colonization success for the “healthy” community were due to the presence of an established “_degraded_” community, or simply to the presence of _an_ established community. The manuscript acknowledges this point in the paragraph at line 299, and makes a reasonable case for focusing on a situation where a degraded community has already established. Nonetheless, the discussion here felt a bit superficial.

Response: This is a great point, and a limitation of the study which we are keen to acknowledge. In the paragraph at line 330ff we have made edits to make the conclusions more cautious, and to further acknowledge this limitation. We have also further explained the case for focussing on this particular degraded community scenario.

I was left wondering what the community ecology literature says about colonization or invasion of established communities, and how to interpret the results here in light of that literature. As one small piece of a discussion of this issue, it might be valuable to consider the role of disturbance in initiating new succession sequences; could something like a flood disturb a degraded community to the extent that it then became invasible by the healthy community?

Response: We think so, and we're excited that you've made this logical connection! The idea that disturbance might be used to destabilise a degraded community, breaking its biotic monopoly to facilitate restoration, is the focus of the final chapter of the PhD thesis associated with this work. We have undertaken significant further work which is currently in prep/under review to bridge the gap between this study and that conclusion, so we're reluctant to delve too deep into that hypothesis in this manuscript without being able to present the underpinning logic!

Similarly, if it is in fact the degraded community that is important, it is not clear to what extent the results were attributable to the presence of a degraded community in general, rather than to the presence of *P. antipodarum* in particular. My understanding is that this species is a particularly successful and impactful invader of stream ecosystems, including perhaps those where abiotic conditions have not been degraded. The reader wonders to what extent the results represent the uniquely strong effects of this species, rather than a generalizable effect of degraded communities.

Response: Yes, this is a great point. It is possible that these results reflect *P. antipodarum* being a strong interactor rather than a general effect of a degraded community. Nevertheless, because this snail is so prominent in degraded communities both in New Zealand and further afield it serves as a great example of what could happen. We now acknowledge this possibility around line 332. Although further studies will be needed to determine if there are general effects of degraded communities in restoration contexts, by demonstrating that such effects can occur we hope our study will stimulate that work.

We do wonder whether invasive ability might actually covary with the negative resistance and resilience as species displays once established, but that is another question beyond the scope of this work.

Reviewer 1 points out that the degraded community colonist, *P. antipodarum*, might outcompete the healthy colonist community regardless of the order in which the two arrive. This hypothesis cannot be evaluated given the study design, of course, but it does seem plausible given the success of *P. antipodarum* as an invader, and it calls into question the framing of the results in terms of priority effects. Fortunately, even in this case the paper seems significant, because the message that biotic interactions can hinder restoration goals even when abiotic conditions are restored seems like an important and novel one, and it still stands. Nonetheless, the reviewer's comment emphasizes the need to better justify, tone down, or abandon the framing in terms of priority effects. If you do retain

this framing, the reviewer offers some important suggestions about defining this concept more clearly and with a stronger link to existing literature.

Response: We have decided to retain the priority effects framing, but have made significant edits (detailed below) to address this.

Both reviewers also raise critiques of your use of the resistance and resilience concepts, highlighting some ways in which the manuscript falls short of fully engaging with these ideas.

Response: Yes, we did not utilise these concepts fully, and we have addressed the reviewers' concerns below.

Beyond these major conceptual concerns, the reviewers raise a number of suggestions and questions which should be addressed in a revised manuscript. Some of these bear directly on the strength of the data; for instance, Reviewer 2 asks whether you were able to accurately determine that subimagos had emerged from the mesocosm where you found them resting, rather than from another mesocosm. To their detailed comments I'll add one of my own: I was a little confused by the *Deleatidium* results (paragraph at line 258): how is it that high abundance of this species in the degraded treatment at the end of the experiment is consistent with higher emergence of this species from the degraded treatment?

Response: Good question! We should have made this clearer, and have added more explanation to the results (line 296ff). It is likely that more individuals emerged from the control mesocosms between days 6 and 42 than the degraded mesocosms, due to the stunted development of *Deleatidium* in the presence of a degraded community. Therefore, while initial emergence from the degraded mesocosms was higher, the remaining mayflies which were too small to emerge initially were then delayed by degraded community presence (e.g. due to lack of resources required for growth/development).

Referee: 1

Comments to the Author(s)

This manuscript points out that while many studies and discussions seek to promote ecosystem resiliency, degraded ecosystems can also be resistant or resilient in ways that prevent their recovery to previous, undisturbed states. While environmental filtering is an important determinant of community composition and structure, and can be a barrier to restoration, this study instead looks at the effect of biotic interactions on recolonization of restored habitat. Overall, their finding is that the presence of species that were adapted to degraded conditions can hinder community recovery even once conditions improve. This is an important message. This manuscript was well-written and an enjoyable read – I appreciated how the authors connected this very practical question (how to restore communities) with ecological theory. The discussion of potential longer-term consequences through effects on fitness is also very nice.

Having said that, I think that the paper would benefit from a more clear definition of priority effects. Simply being there first does not imply that the reason a species/guild/community can persist is due to priority effects – they may be competitively superior anyway, in which case it wouldn't matter if they had arrived first. Since *P. antipodarum* is an invasive species worldwide, could it establish even in healthy stream communities in the study area? There are several ways to define priority effects – two through niche theory (niche preemption or niche modification), and one through coexistence theory (that interspecific competition is stronger than intraspecific competition, so whichever competitor has the higher abundance will have the upperhand). I could imagine any of these three mechanisms being true for the authors' study system (the snails could certainly modify available niches, and L340-341 suggests that they may be fairly impervious to intraspecific competition). It is unfortunate that the authors did not have a treatment with high abundance of the "original" community and low/medium abundance of the "degraded" community, or some other combination of relative densities – this would be the way to establish whether a priority effect is truly occurring. Regardless, the authors should be more clear about exactly what they mean by priority effects, if they wish to use it as a central explanation of their findings.

Response: This is great feedback, thank you! We have chosen to retain the priority effects framing of the paper as we feel these concepts may be usefully applied (if unconventionally) in the restoration space. However, we appreciate these ideas have not been well developed here, thus we have made significant edits to better explain our thinking. Definitions have been clarified (line 73), and additional insights added to the discussion (e.g. lines 380, 398, 402). We have also pointed out the limitations of using a *P. antipodarum*-dominated degraded community to infer generalised degraded community effects (line 332).

My other major comment is that this paper is framed (including in the title) as being about resistance and resilience, yet does not perform any of the analyses typically associated with these questions. By excluding the "degraded" community from the ordination analyses, the authors don't really look at whether they are resistant/resilient to the colonizers' arrival. Instead of characterizing the total community, most of the analyses are about the colonizer community. This is definitely interesting, and required in order to understand the mechanisms happening. However, it is just a bit of a surprise given the main framing of the paper. This could be addressed by making some more specific links in the discussion – for example, somewhere in the discussion, maybe around lines 301-302, say something like, "The degraded community was indeed very resistant to change in community composition, as shown by the maintenance and even increase of *P. antipodarum* abundance after the addition of the colonist community."

Response: We agree – we definitely could have engaged with the resistance and resilience terms more substantially. A sentence has been added as suggested (line 330), and links to negative resistance and resilience have been made clearer (e.g. line 335).

Addressing these issues would require just some editing and potentially a slight reframing of parts of the introduction and discussion. This is an excellent study and manuscript, and with some clarification it will be an important contribution to both conceptual and applied ecology literature. It is also very nice to see the authors bringing ideas that are familiar in terrestrial settings into the aquatic realm; such connections are always beneficial.

I have few other comments. The description of the experiment is thorough and the design seems well-thought-out, and statistical analyses appear to be appropriate.

L305-306: As mentioned in the general comment above, this experiment doesn't exactly prove that priority effects are the mechanism through which this is occurring. Changing wording to "suggests" could be better.

Response: Wording altered accordingly.

L306-308: Likewise, here I would suggest changing the wording to "existing communities and any historical contingencies" would be more appropriate.

Response: Wording altered accordingly.

Referee: 2

Comments to the Author(s)

Title

I suggest to add a word like “aquatic” or “stream” to the title to underpin that the study was conducted in the aquatic environment and focused on stream restoration.

Response: Title has been updated to more accurately reflect the study system.

Abstract

L20-23. This sentence, definition and prove of the very important term “negative resistance and resilience” is one of the main conclusions of the paper, thus it would fit much better at the end of the abstract as a conclusion rather than as the second sentence.

Response: This is a very helpful insight from fresh eyes! ‘Negative’ has been removed from the start of the abstract and emphasised in the final sentence.

Introduction

L68 I think it is important to mention that this reference refers to terrestrial plant communities and it is not one on one transferable to aquatic flowing (not standing) systems.

Response: Updated for clarity.

L79ff This is fine for me, but isn’t there a time factor inherent to the “succession”?

Response: There is! I have now addressed time more specifically in the discussion (line 403).

L105 the accessory sentence could be deleted, as the species are not lost to the community in general if they emerge, they are only lost to the current aquatic spot.

Response: Sentence removed.

Methods

L181 asses needs one more “s”

Response: Corrected.

L208ff The authors need to explain how they ensured that the counted subimagines emerged from the specific mesocosm and not from an adjacent one of a potential different treatment.

Response: Sentence added acknowledging the poor flying ability of subimagos, and therefore low likelihood of movement between mesocosms (line 216).

Why weren’t the imagines counts also done in week 2 (day 13) and the end (42)?

Response: On day 13, it was decided that counts would not be carried out because weather conditions would have made these unreliable. Higher winds were blowing imagos away soon after emergence, so counts would have been misleading. On day 42, no imagos were observed. The onset of colder temperatures leading into winter had likely halted emergence: *Deleatidium* generally exhibit continuous growth and development except for cold torpor during winter (see Hury, 1996). In an ideal world, the experiment would have begun a few months earlier, however we think that the emergence data presented in combination with the development data from the end of the experiment show an interesting and important response to degraded community presence.

Reference:

Huryn, D. (1996). Temperature- dependent growth and life cycle of *Deleatidium* (Ephemeroptera: Leptophlebiidae) in two high- country streams in New Zealand. *Freshwater Biology*, 36, 351-361.

Results

L238ff The text jumps between results of figure 2 and 3. I suggest to first explain figure 2 and then add results of figure 3.

Response: Thanks for this feedback! We wondered whether the previous order made the logic easier to follow. The order has now been altered to describe these figures separately, hopefully still making clear the links between these results (line 247ff).

The explained variance of the two NMDS axes need to be added, to understand their influence.

Response: In contrast to other Principal Component Methods, there is no variance associated with each axis in NMDS (Legendre and Legendre 2012). However, we have reported ordination stress in Figure 3 which represents the difference between distances in the reduced dimensions of the NMDS ordination and in the complete multidimensional space, thus is a good indication of ordination fit.

Legendre, P. and L. Legendre (2012). Nonmetric multidimensional scaling (nMDS). Numerical Ecology, Elsevier: 513.

L243 the authors should add if the loss of those species was due to natural emergence, drift, predation or other factors

Response: Sentence added to clarify this was likely due to emergence or drift (line 269).

Pool and riffle samples of each sample date were kept separately (see methods), so the results should be indicated or at least indicated if there were no significant differences.

Response: This sampling method was designed to give more representative values of taxon abundances in the mesocosms, taking into account potential differences between habitats. Therefore, we do not think combining them invalidates any results. Furthermore, the specific influence of habitat is not relevant to our hypotheses, thus we don't think separating out the habitat types adds to the story.

For interest's sake, there was no difference in *Potamopyrgus* abundances between habitats, but abundances of other taxa did differ between riffles and pools, with more in the riffle habitat (as would be expected). There was no interaction between mesocosm and habitat type.

Discussion

L297 "is" important

Response: Amendment made.

L305/306 please re-write. The conclusion is too straight forward and does not account for any time frame longer than 13 days. Even the results of the end day of the experiment show that there will be some succession, not to mention potential changes in upcoming years or decades.

Response: Sentence has been edited to be a little more cautious. Time frames have now been more specifically addressed in line 403ff.

L307 "likely" can be deleted

Response: Removed.

L327 Just a comment: particularly on the long run – how long will this negative resistance remain, will be one of the next questions to be answered

Response: Couldn't agree more! Though this relies on reliable, long-term monitoring which is painfully difficult to achieve!

L331ff this is true for the grazers and collectors but what about the predators?

Response: Good point! Sentence added at the end of this paragraph noting implications for food webs (line 385).

L347 again I think this cannot be stated in an absolute manner; there are no data yet that recovery is completely prevented (only 13/42 days). Recovery is probably delayed and most probably delayed for years but not forever.

Response: Wording altered to be more cautious.

L378 impede or delay or hinder is in the sense above better than “prevent”

Response: Agreed. Wording altered accordingly.

L382ff this is a very important point and might also be stated in the abstract

Response: Agreed, however because of the tight word limit, we're reluctant to remove other information already included in the abstract to further explain this point there.

Figures

Fig. 2. Normally the results of the end day should be displayed together with the results of the first three samplings in one diagram. Thus, B, D and F is not necessary. The curves probably do not look so nice because they increase again but the message is still the same and the interpretation in the result section can remain.

Response: Thanks for this feedback. The figure has been edited to combine these two. We have retained the statistical analysis methods to account for the differing methods during sampling (counting invertebrates by eye versus under a microscope), but have included both models on one plot.

Fig 4. It should be added why there are no counts of stage 3 in Fig 4 A.

Response: We have added some clarifications to the results paragraph (line 296ff). There were fewer individuals in the control mesocosms in general, which happened to not include any at stage 3.

I am not sure but someone has to decide if the plural of subimago is subimagines, subimagos or subimagoes. The latter one is wrong I think.

Response: Changed to subimagos.

And finally a comment of our own: we have corrected use of the term ‘positive feedback’ to fit in with the accepted definition of these terms in systems theory, where negative forces are stabilising, and positive forces are transformative.

Appendix B

Response to reviewers:

Associate Editor

Comments to Author:

Thank you for your careful attention to revising the manuscript and responding to the reviews. The manuscript is now close to being acceptable for publication in Proc B. Please attend to the few remaining minor issues highlighted in the reviewers' comments. Also, please make sure that the data are fully accessible - one of the reviewers noted that, while they were able to find the data via the BioHeritage website, the link provided in the paper didn't work for them.

We have followed this access issue up with the BioHeritage repository, and the link has now been fixed. Apologies for this!

Please address these issues expeditiously by submitting a revised version. This revision will not be sent out for further review, but will be checked by our editorial staff to make sure that the paper is suitable to be sent to production.

Thank you again for submitting this interesting work to Proc B, we look forward to seeing it in print.

Reviewer(s)' Comments to Author:

Referee: 1

Comments to the Author(s).

This manuscript describes a well-designed study to investigate the effects of biotic interactions on the establishment of natural invertebrate communities to restored stream habitats. The authors comprehensively addressed most comments from the previous reviews, adding precision and nuance to the introduction and discussion as well as explaining a few methodological issues. I consider this to be a strong and well-written manuscript and have no major comments.

Minor comments:

L179 and Figure 3: The methods text states that the snails were left out of community analyses, but the NMDS plots show species 21 as *Potamopyrgus*. This is a little confusing.

So grateful for the reviewers identifying this typo! This error has been fixed in the caption – *Potomopyrgus* should not have been included, hence there is no taxon 29 (as pointed out by the second referee).

Figure 2: I can see from the documents included that Figure 2 was edited according to suggestions by a previous reviewer. However, I find this presentation with a smooth line for days 1-13 and then some points at day 42 to be very jarring. I don't consider it to be an improvement on the previous Figure 2, but it is ultimately up to the authors' preference!

We are inclined to agree, however have decided to stick with the revised figure as this better reflects changes over time.

Referee: 2

Comments to the Author(s).

After all the very good improvements I have only one small comment: in Figure 3 the species number 29 is missing in the figure. This should be added.

Comment addressed above.